# Teaching Environmental Themes within the "Scientific Awakening" Course in Moroccan Primary School: Approaches, Methods and Difficulties

Bouchta El Batri [1,2], Lhoussaine Maskour [3,4,*], Jamal Ksiksou [5], Eila Jeronen [6], Jalal Ismaili [7], Anouar Alami [2] and Mohammed Lachkar [2]

1   Regional Center for Education and Training Professions (CRMEF Fez-Meknes), Fez 30000, Morocco
2   Engineering Laboratory of Organometallic, Molecular Materials, and Environment (LIMOME), Sidi Mohammed Ben Abdellah University, Fez 30000, Morocco
3   Regional Center for Education and Training Professions (CRMEF), Dakhla-Oued Eddahab 73000, Morocco
4   Laboratory of Science and Technology Research (LRST), ESEF, Ibn Zohr University, Agadir 80000, Morocco
5   Sociology-Psychology Laboratory, Faculty of Letters and Human Sciences Dhar El Mehraz, Sidi Mohamed Ben Abdellah University, Fez 30000, Morocco
6   Faculty of Education, University of Oulu, 90014 Oulu, Finland
7   Laboratory of Humanities, Entrepreneurship and Digital Studies, Moulay Ismail University of Meknes, Meknes 50050, Morocco
*   Correspondence: lhomaskour@gmail.com

**Abstract:** In addition to identifying the pedagogical approaches favoured by teachers in environmental education, the study aims to reveal the impact of the teaching methods and tools used as well as the teaching difficulties encountered on the effectiveness of teachers' pedagogical action, particularly in the "Scientific Awakening" course.The study concerns a sample of 636 primary school teachers from the urban and rural areas of the Fez-Meknes Regional Academy of Education in Morocco. The data was collected using a 37-item questionnaire covering the following variables: the teaching methods adopted, the preferred pedagogical approaches, the teaching material used, and the teaching difficulties encountered. The study fits under a descriptive correlational design. The most used teaching methods were the teacher-centred oral methods (dialogue and demonstration method) lacking learner-centred activities (working in small groups, discovery method). The least used ones were laboratory experiments and ICT-based demonstrations. The study shows that teachers who use active methods are the most aware of difficulties in teaching environmental issues and were the most capable of effectively solving learning problems and achieving pedagogical objectives. The study shows that Moroccan primary school teachers need in-service training for the adoption of a systemic and interdisciplinary pedagogical approach. In addition, to address the issue of the alarming failure to complete the "Scientific Awakening" program, we recommend continuous training for the benefit of teachers. This training should cover the appropriate active methods to effectively complete this program. Finally, the study underlined the need to solve the problem of the enormous lack of teaching tools. Certainly, this shortage significantly influences the pedagogical action of teachers regardless of the pedagogical methods adopted.

**Keywords:** environmental education; primary education teachers; scientific awakening; pedagogical approach; teaching methods; teaching effectiveness; descriptive correlational design

## 1. Introduction

Currently, it is acknowledged that human activities are the cause of several environmental problems, whether on a national or global scale. Some of these problems may reach serious irreversible thresholds. In Morocco, we can mention, as an example, the continuous regression of biodiversity, forests, and the degradation of water in quantity and quality [1–3].

Moreover, we daily witness behaviours that reveal a poor familiarity with environmental education, increased individualism, and a quasi-complete loss of the sense of common good. Environmental education is an essential component of basic education. It relates to our relationship with the environment; one of the bases of personal and social development.

Sauvé [4] proposed a definition of environmental education that incorporates practical elements. This definition is accepted and widely reported by the research community [5,6] and others, and it states:

Environmental education (EE) is an integral dimension of the development of individuals and social groups, which concerns their relationship to the environment. Beyond the simple transmission of knowledge, it favours the construction of collective knowledge from a critical perspective. It aims to develop useful know-how associated with real skills. It calls for the development of an environmental ethic and the adoption of attitudes, values, and behaviour imbued with this ethic. It favours cooperative learning in, by, and for environmental action [4]. Asano and Poletto [7] have shown that environmental education is fundamental to raise awareness so that natural resources are used in a sustainable way by current and future generations, promoting changes in habits and attitudes, creating a balance between society and nature.

Environmental education should be seen as a strategy for solving environmental problems, as well as a process for the development of individuals and social groups [8]. It contributes to the development of the individual in his relations with the environment, particularly, the development of critical thinking of individuals vis-à-vis their own relations with the environment. Environmental Education also aims to undertake sustainable, more balanced, and more ecological relationships with the environment without domination or overexploitation. It plays a very important role in the construction of new values, attitudes, and behaviours on the part of people and social groups [9–14]. Environmental education also contributes to strengthening social relations and developing a sense of belonging through individual and collective actions in favour of the environment [15]. This makes it possible to effectively solve certain environmental problems and, therefore, improve the quality of life [16]. Some authors even go even further by discussing environmental and educational justification [17].

Regarding the pedagogical dimension, environmental education has fostered new opportunities for the educational act, both in terms of systemic and critical thinking skills [18,19], interdisciplinary content [20–22], and contextual content [22,23]. An equivalent shift took place at the methodological level by favouring interactive and collaborative action [20,24,25] or at the level of ethical development [26,27] and behaviour [9,13,28]. Environmental education is considered a cross-cutting theme closely linked to the natural sciences [29]. According to Dias [30], the environment is constantly changing due to changes in its biotic and abiotic components including positive or negative anthropogenic actions. Accordingly, understanding knowledge about biology and the environment is important for the development of environmental education [20].

The "Scientific Awakening" program, around which this research revolves, is a multidisciplinary program rich with environmental themes. It includes areas relative to life and earth sciences, ecology, physics, chemistry, and astronomy. However, in addition to the development of knowledge, this program contains areas relating to awareness of environment-friendly attitudes and behaviours as well as contributions to the preservation of the local environment. We have identified this particularly in the following areas:

- Water pollution due to human activities.
- Raising awareness on the importance of water conservation.
- The contribution to the preservation of the forest.
- How to protect the soil, etc.

We can, therefore, say that the "Scientific Awakening" programme, if implemented with appropriate active and creative methods, can develop not only the learner's knowledge relating to the environment, but also attitudes, behaviours and even individual and collective action skills that are in favour of the environment and well-being.

Many studies dealing with the teaching and learning of environmental issues at the elementary level have shown that a considerable number of teachers have problems and teaching difficulties in teaching these issues [31–37]. The reason for the teaching difficulties is considered to be insufficient education concerning environmental issues [31,34,35,38] and support of administration [31,35,37,39]. According to previous studies, teachers also have cognitive, methodological, and moral deficits. Cognitive deficits are related to a lack of knowledge [31,34,36,38,40] e.g., about ecological concepts and processes which concern relationships between nature and human beings [41,42]. The methodological deficits are linked to the pedagogical approaches concerning the teaching themes and teachers' professional development [43–49]. Moral deficits again are linked to the lack of motivation and professional exhaustion of teachers [35,37,39,50].

In addition, some researchers have mentioned other pedagogical problems which can significantly influence the quality of learning such as the content of environment education programmes [32,51], the characteristics of the students [32,35,52], the lack of educational material [31,35,37,53] and the lack of time allocated to certain units of the school curriculum [35,37]. This study investigated deficiencies and their effects on the teaching of environmental education in Moroccan primary education studying the pedagogical approaches and teaching methods preferred by primary education teachers. To the best of our knowledge, little previous research on this point of view has been published.

In characterising the pedagogical approaches preferred by the Moroccan primary education teachers in environmental education, the pedagogical approaches mentioned by Sauvé [54,55], Sauvé, Villemagne and Orellana [56], and Villemagne et al. [57] were used. These pedagogical approaches are related to the learning objectives (cognitive, affective, spiritual (religious), moral, behaviourist, pragmatic, and praxis) and the learning processes (experiential, systemic, holistic, interdisciplinary, cooperative, critical, and the problem-based approach (problem-solving pedagogy)).

In the previous studies, e.g., the following pedagogical approaches to environmental education have been examined in different contexts: affective approach [58–62], religious approach [63–65], behaviourist approach [66,67], cooperative approach [68,69] and approach by solving environmental problems [70,71].

The preferred teaching methods for Moroccan primary education teachers were investigated by using the typology of El Batri et al. [49] in which teaching methods are divided into traditional, transmissive teacher-centred methods (dogmatic and interrogative ones) and active student-centred methods (active dialogue, work in small groups, discovery learning, Prior preparations of students, demonstrative method, and laboratory experiments).

In life and earth sciences, the frequent use of several active methods [25,72] and the reasoned diversification of teaching methods can be considered as positive indicators of effective learning [49,73]. With the student-centred teaching methods, students' environmental awareness and environmental perceptions can be fostered [61,62,74–76]. Using repeated nature experiences, on the other hand, the development of an emphatic relationship with nature can be supported [58,61,62,74,77,78]. According to Otto and Pensini [79], environmental knowledge and connection to nature are two complementary drivers of ecological behavior.

The repeated use of traditional transmissive methods together with one teaching tool (course textbook) again is an indicator of less effective learning [49,73]. This kind of routine learning can also lead to students accepting any fixed knowledge. It deters students from becoming independent and innovative critical thinkers and decision-makers, missing all vital components of environmental education [62].

## 2. Research Aim and Questions

Environmental education can be seen as a lifelong learning process [80], which itself is as much teaching as learning and requires an understanding of both environmental education theories and practices. Its mission statement is to increase environmental awareness,

develop skills to acquire and use information, develop environment-friendly attitudes, and enable participation [81]. Therefore, and from the perspective of environmental education in elementary education, an important question raises concerning the nature of deficiencies in teaching and how they affect learning. This study investigated the shortcomings and their effects on the teaching of environmental education in Moroccan primary education by examining the pedagogical approaches and teaching methods preferred by primary education teachers. To our knowledge, little previous research has been published from this point of view.

This study answers three research questions:

- What are the preferred pedagogical approaches, the used teaching methods and tools and the teaching difficulties encountered by Moroccan primary school teachers in environmental education, particularly in the "Scientific Awakening" course?
- Is there a relationship between the teaching methods used and the teaching difficulties encountered by teachers in their teaching of the "Scientific Awakening" program?
- To what extent do the teaching methods used, and the teaching difficulties encountered by teachers influence the effectiveness of teaching relating to the environmental themes of the "Scientific Awakening" course?

The results are useful for the development of both the curricula/instruction of environmental education in elementary and the teacher training not only in Morocco but also in other countries.

## 3. Materials and Methods

### 3.1. Research Design

First, the teachers' favourite pedagogical approaches, the teaching methods and tools used, and the teaching difficulties faced by teachers were described and characterised using a descriptive design. However, upon performing a statistical analysis, significant correlations were found between these variables; initially, they were found between the teaching methods used by the teachers and the teaching difficulties they encountered and also between these teaching methods and teaching difficulties and the effectiveness of teaching. Thus, in the description of the results, the teaching difficulties encountered by the teachers are described in more detail in the context using a descriptive correlational design. This design is most appropriate for this study because it allows the data and correlations between the variables studied to be described, and in the light of this description, the plausible explanation is determined [82,83]. The correlations found between the variables studied do not necessarily imply causality [84,85] and they do not exclude alternative explanations for a correlation between two variables [85]. This same design (descriptive correlational) has also been used in recent studies [32,83,86].

### 3.2. Sample

The sampling method used is of the stratified character. This is to ensure that the sample is representative of both urban and rural areas. The educational and socioeconomic differences existing between the two environments have been considered also in this study. These differences have been revealed by very recent studies carried out in Morocco and in the same region as Fès-Meknes [32,86,87]. Within each stratum (middle), we took an exhaustive sample. The population targeted by the study concerns Arabised teachers of the second primary cycle (4th, 5th, and 6th year of primary education) and particularly teachers of the "Scientific Awakening" course. The study targeted 636 teachers, 401 of whom were from urban areas and 235 from rural ones. The syllabus of the "Scientific Awakening" course contains lessons related to physical science and others related to life and earth sciences (LES). This course is taught for one hour and half per week over two sessions. For Physics, we find the following themes: electricity, transformations of matter, heat exchange, movement, gases, energy, light, pressure, astronomy, solubility, and mixtures and separation of their components.

The life and earth sciences (LES) part of the syllabus is rich in content related to the environment and environmental education. Table 1 illustrates the main axes relating to the LESs of this syllabus.

**Table 1.** Main axes relating to the LESs of the "Scientific Awakening" syllabus.

| School Level | Main Axes of the Syllabus |
|---|---|
| Fourth year primary | - Nutrition and digestion<br>- The life cycle of an animal and a plant<br>- The classification of vertebrates<br>- Human use of water<br>- The water pollution<br>- Awareness of the importance of water conservation<br>- The identification of living beings in nature |
| Fifth year primary | - The interrelationships between the components of the forest<br>- Raising awareness of the importance of the forest environment<br>- The contribution to the preservation of the forest<br>- The forest as a source of energy<br>- Comparison of the characteristics of herbivorous and carnivorous animals<br>- Diagram of food chains and food webs |
| Sixth grade primary | - Identification of soil components<br>- Soil erosion factors and how to protect it<br>- The role of soil in agriculture |

Primary teachers are not all sufficiently trained to teach all the themes of this course, especially since they come from very different university backgrounds. In this context, it should be noted that the initial training of Moroccan primary school teachers is very heterogeneous. Some have a Bachelor of Science, others have a Bachelor of Arts in literary, economic, or even vocational studies. Apart from the teachers who have undergone initial scientific training, the others do not have in-depth training on the environmental issues included in the "Scientific Awakening" program. In addition, the six months they spend in the "regional centres for education and training professions" to qualify for the teaching profession are insufficient to mend all the cognitive and theoretical gaps. This is because two thirds of the time reserved for this professional training is devoted exclusively to the practical and educational aspects (internships).

Table 2 shows the quantitative data for the selected sample.

**Table 2.** Quantitative data concerning the sample studied.

| | Total Number of Teachers (Arabized) | Sample Studied *N* | % |
|---|---|---|---|
| Urban environment (Direction of Fez) | 734 | 401 | 54.63 |
| Rural environment (Direction of Moulay Yacoub) | 326 | 235 | 72 |
| Total | 1060 | 636 | 60 |

Table 3 shows the demographic data for the sample studied, especially age and gender.

We found that most teachers in urban areas are older in age and are females. In fact, 67.58% of urban teachers are over 49 years old. While teachers in rural areas are younger and those aged over 49 represent only 42.55 in rural areas. The female gender represents 63.59% in urban areas and only 49.36 in rural areas.

**Table 3.** Demographic data of the sample studied (age and gender).

| | | Gender | | | | | | Total | |
| | | No Response | | Male | | Female | | | |
| | | N | % | N | % | N | % | N | % |
|---|---|---|---|---|---|---|---|---|---|
| Teacher's age | No response | 12 | 1.88 | 2 | 0.31 | 2 | 0.31 | 16 | 2.5 |
| | Age < 35 years old | 0 | 0 | 1 | 0.15 | 48 | 7.54 | 49 | 7.7 |
| | 35 ≤ Age ≤ 49 | 0 | 0 | 59 | 9.27 | 141 | 22.16 | 200 | 31.44 |
| | Age > 49 years old | 0 | 0 | 191 | 30 | 180 | 28.30 | 371 | 58.33 |
| | Total | 12 | 1.88 | 253 | 39.77 | 371 | 58.33 | 636 | 100 |

*3.3. Data Collection Tool*

To answer the research questions, we used a four-component questionnaire. Apart from essential general information (age, gender, institution, county, and delegation), the questionnaire focused on four fundamental parts: the adopted teaching methods (10 items), the teaching tools used (7 items), the teaching difficulties encountered (8 items), and the pedagogical approaches (12 items) favoured by Moroccan primary school teachers, particularly for the "Scientific Awakening" course and the environmental themes which characterise it.

Almost the same teaching methods as those defined by El Batri et al. [49] were included in the questionnaire. The ones included were the dogmatic method, the interrogative method, the active dialogue, working in small groups, the discovery method, the demonstrative method, and laboratory experiments (Table 4). To assess the frequency of effective use of each teaching method, the five-point Likert scale was used (0 points to "no response", 1 point to "not used", 2 points to "rarely", 3 to "occasionally" and 4 points to "often").

**Table 4.** Extract of the teaching methods included into the questionnaire.

| |
|---|
| Dogmatic method: The teacher explains the lesson and the students listen without participation |
| Interrogation-response method: Partial participation of students by answering some questions asked by the teacher |
| Dialogue method with active participation of students at all stages of the lesson |
| Group work method for performing works and assignments either inside or outside the classroom. |
| Discovery Method: Encourage the students to learn through their own explorations, experiments and research around an identified problem. |
| Realisation of laboratory experiments. |

The teaching tools used by the teachers for the "Scientific Awakening" course were mapped with the help of various tools, probably also used in the course in question, such as textbook, documents prepared by the teacher, information and communication technology (ICT), laboratory tools, fresh material, field trips, etc. Also, in this second part of the questionnaire, a five-point Likert scale was used (0 points to "no response", 1 point to "not used", 2 points to "rarely", 3 to "occasionally" and 4 points to "often").

As for the teaching difficulties, the various challenges usually encountered by teachers and especially presented in the literature were listed such as teaching difficulties related to:

- the content of the taught program [32,51],
- the characteristics of the students [32,35,52],
- the lack of educational tools [31,35,37,53],
- the time allocated for teaching [35,37] and

- the extra-large number of students per class [88,89]. It is noted that the usual number of students per class in Moroccan public schools is about 40 [49].

Before giving choices concerning the nature of these difficulties in the questionnaire, we asked two questions: one closed question on the presence or absence of these difficulties and the other multiple-choice question relating to the percentage of completion of the "Scientific Awakening" program (70%, 80%, 90%, 100%).

For pedagogical approaches related to environmental education, we used the typology of pedagogical approaches proposed by Sauvé [55], Sauvé et al. [56], and Villemagne et al. [57]. Table 5 shows an extract of the proposed pedagogical approaches. To characterise the opinions of the teachers on each type of pedagogical approach, the five-point Likert scale was used (0 points to "no answer", 1 point to "disagree", 2 points to "indifferent", 3 points to "somewhat agree" and 4 points to "strongly agree").

**Table 5.** Extract from the pedagogical approaches included into the questionnaire.

| | |
|---|---|
| (1) | The cognitive approach: it aims to transmit a set of environmental knowledge to students. |
| (2) | The religious approach: it is centred on the development of attitudes and values consistent with a religion. |
| (3) | The behaviourist approach aims to adopt an appropriate behaviour with respect to the environment. |
| (4) | The experiential approach aims at upholding direct contact with real-life situations, the realisation of experiments and interaction with the living environment to develop handling skills, on the one hand, and, on the other hand, to understand natural phenomena. |
| (5) | The holistic approach: takes into account all the dimensions of the subject and allows the development of a global vision of socio-environmental and educational realities. |
| (6) | The systemic approach: takes into account the networks of interrelations and interdependencies within the environment (ecosystems). |
| (7) | The interdisciplinary approach: integrates knowledge from different disciplines for better understanding and informed action. |

Concerning the validity of the tool used, for the part relating to pedagogical methods and tools, in addition to our professional experience as a teacher of life and earth sciences, we went through the literature to extract the pedagogical methods and tools likely to be used in the "Scientific Awakening" course. This part has been worked on by four teacher-researchers who combine field experience and scientific research in the field of education. The very low proportion given by the teachers to the section "other methods" (0.3%) and "other tools" (0.5%) shows that the main teaching methods and tools used are those included in the questionnaire. These methods and tools have been validated and used in other recent studies [49,73]. The same goes for pedagogical approaches relating to environmental education. They have been described in several other studies [54,57]. The discriminating power of the instrument was tested a second time six weeks after the first data collection (November 2019). This test was carried out on 15 randomly-chosen teachers. They generally indicated the same teaching methods and tools as well as the same main teaching difficulties that they mentioned during the first data collection. The teachers who "strongly agreed" with the same pedagogical approaches as those reported during the first data collection represent about 87%. This is an important element in the reliability of results. The reliability index (Cronbach's alpha) applied to each part of the questionnaire as well as to all the items gave the following results: 0.693 for teaching methods and tools; 0.742 for teaching difficulties; 0.883 for pedagogical approaches; and 0.818 for all the items of the questionnaire. This indicates that the internal consistency of the questionnaire is satisfactory.

*3.4. Data Collection*

Data collection was administered in person. In fact, one researcher visited almost all the surveyed schools in collaboration with the headmasters of the concerned institutions. During the visits at schools, he had the opportunity to converse with the teachers and to explain the purpose of the questionnaire, the framework of the study, the items, and how to answer the questions. However, in general, the researcher did not register serious problems concerning the understanding of the teachers and their answer to the items of the questionnaire. Some teachers preferred to return the questionnaires the next day. This is to take time to reflect on and answer the questions consistently. The researcher collaborated with the school headmasters to collect the questionnaires. To identify the teaching methods and tools that have a significant impact, in terms of results only, the rates of "often used" methods and tools were kept, i.e., the teaching methods and tools that were declared "occasionally" or "rarely used" were disregarded. Also, at the level of pedagogical approaches, only the opinions of the teachers who "strongly agree" for each type of pedagogical approach were counted.

The ethical principles of research were taken into account throughout the data collection process. First, the necessary permissions were obtained to contact the teachers. The approval to conduct field surveys was delivered by the university, the Regional Academy of Education (Fès-Meknes), and the two provincial education directorates (Fez and Moulay Yacoub). The names of the participants and the schools were kept confidential. It should be noted that some teachers did not answer certain questions such as age (2.5%), gender (1.88%), percentage of program completion (6.1%) and the presence of teaching difficulties (0.8%). Since they are not numerous and our retrieved sample is exhaustive, the data collected was used as it is by indicating the missing information under the label "no answer". The data entry and numerical coding was carried out using the statistical software "IBM SPSS 20".

*3.5. Data Analysis*

The data analysis was performed based on descriptive statistics and significant correlations detected between some of the studied variables. This was done through the "IBM SPSS 20" statistical software. The graphs were retrieved using "Excel" software. The descriptive statistics allowed formulating a clear idea about the favourite pedagogical approaches, the frequently used methods and tools and the main teaching difficulties encountered by the teachers. Also, the significant correlations that were identified between some of the variables studied (methods/difficulties and difficulties/% of program completion), allowed us to better explain the impact of these factors (variables) on the effectiveness of teachers' pedagogical action.

## 4. Results

### 4.1. Pedagogical Approaches, Methods and Tools

The pedagogical approaches which the teachers most valued in environmental education were affective-moral, religious, behavioural, cooperative, and problem-solving approaches. Other pedagogical approaches, especially the systemic, interdisciplinary, and critical approach, were less valued (Figure 1).

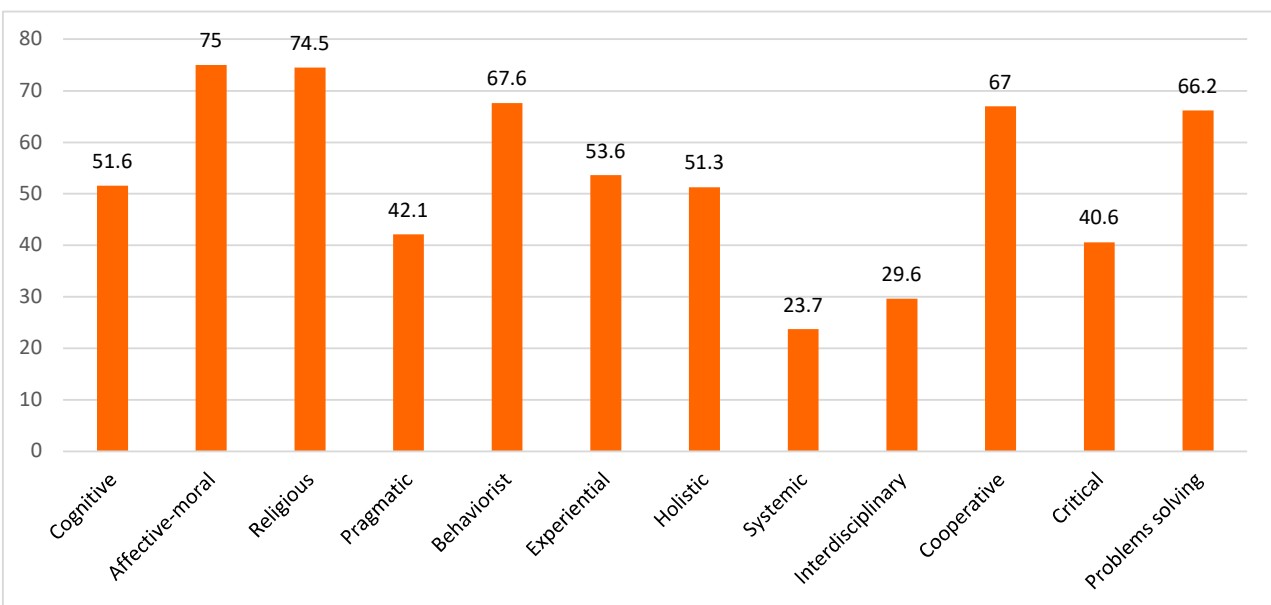

**Figure 1.** Percentage of the "strongly agree" views for each type of the pedagogical approach to environmental education.

Table 6 shows that there is no significant difference between teachers in urban and rural milieus in terms of their preferred approaches.

**Table 6.** Percentages of teachers who totally agree with each type of approach in the two settings (urban and rural).

| Approaches | Environment | | | |
|---|---|---|---|---|
| | Urban (401 Teacher) | | Rural (235 Teacher) | |
| | N | % | N | % |
| Cognitive | 211 | 52.61 | 117 | 49.78 |
| Affective-moral | 301 | 75.06 | 176 | 74.89 |
| Religious | 301 | 75.06 | 173 | 73.61 |
| Pragmatic | 174 | 43.39 | 94 | 40 |
| Behaviourist | 272 | 67.83 | 158 | 67.23 |
| Experiential | 201 | 50.12 | 140 | 59.57 |
| Holistic | 211 | 52.61 | 115 | 48.93 |
| Systemic | 93 | 23.19 | 58 | 24.68 |
| Interdisciplinary | 120 | 29.92 | 68 | 28.93 |
| Cooperative | 276 | 68.82 | 150 | 63.82 |
| Critical | 167 | 41.64 | 91 | 38.72 |
| Problems solving | 264 | 65.83 | 157 | 66.80 |

Figure 2 shows that dialogue and the demonstrative method are the two mostly used methods by the vast majority of the teachers. The school textbook is the teaching tool most often used by almost all teachers (Figure 3).

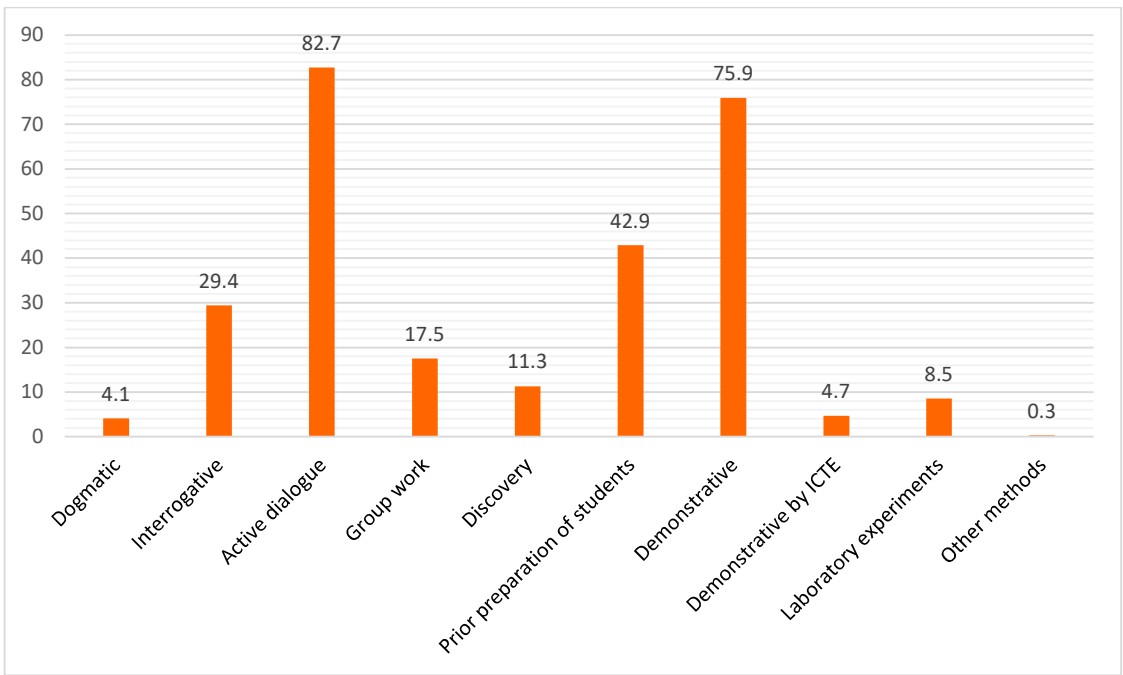

**Figure 2.** Percentage of "frequent" use of each teaching method.

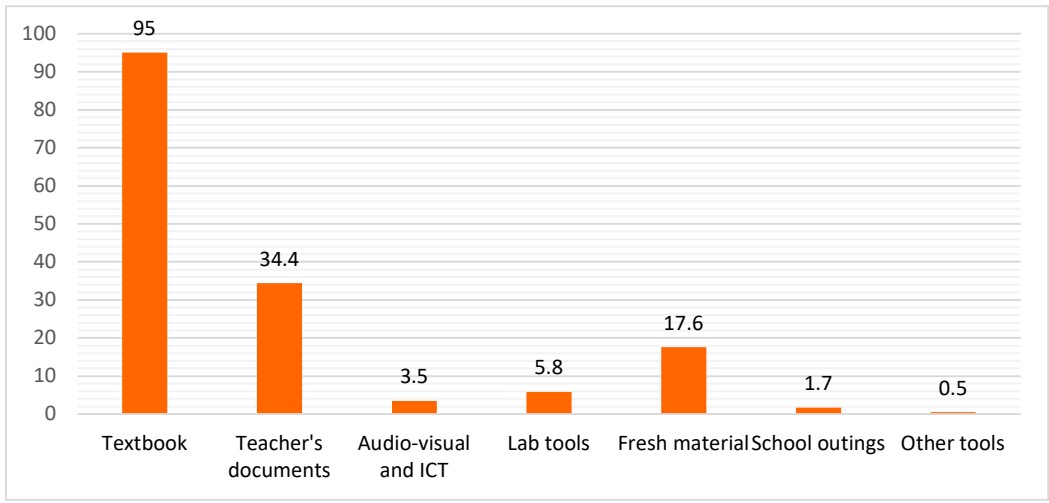

**Figure 3.** Percentage of "frequent" use of each teaching tool.

*4.2. Realisation Rate of the "Scientific Awakening" Programme and Teaching Difficulties*

47.5% of the teachers did not finish the "Scientific Awakening" syllabus (Table 7). This indicates the existence of a real problem concerning the implementation of this program. This is confirmed by 70.9% of teachers who reported having teaching difficulties in teaching this program (Table 8). The teaching difficulties that the teachers reported the most were the lack of didactic tools, the extra-large number of students per class, and the students' poor level of knowledge (Figure 4). The overcrowding problem was more pronounced in urban areas (64.58%) than in rural areas (23.4%) (Table 9). We also note that the complete achievement (100%) of the "Scientific Awakening" program is greater in urban areas (56.1%) than in rural areas (46.38%) (Table 10). In addition, teacher documents, as often used teaching tools, are better reported in urban areas (38.90%) than in rural areas (26.80%) (Table 11). It seems that teachers in rural areas encounter more constraints in their pedagogical work compared with their colleagues in urban areas.

**Table 7.** Percentage of completion of the "Scientific Awakening" programme.

|  | **Frequency** | **Percent** | **Cumulative Percent** |
|---|---|---|---|
| No answer | 39 | 6.1 | 6.1 |
| 70% | 10 | 1.6 | 7.7 |
| 80% | 52 | 8.2 | 15.9 |
| 90% | 201 | 31.6 | 47.5 |
| 100% | 334 | 52.5 | 100 |
| Total | 636 | 100 | |

**Table 8.** Presence of teaching difficulties in environmental subjects in the "Scientific Awakening" course.

|  | **Frequency** | **Percent** |
|---|---|---|
| No answer | 5 | 0.8 |
| Yes | 451 | 70.9 |
| No | 180 | 28.3 |
| Total | 636 | 100 |

**Table 9.** Difficulties relating to the excess number of students per class depending on the environment.

|  |  | **Environment** | | **Total** |
|---|---|---|---|---|
|  |  | **Urban** | **Rural** | |
| Difficulties relating to the excess number of students per class | Yes | 259 (64.58%) | 55 (23.40%) | 314 |
| | No | 142 | 180 | 322 |
| **Total** | | 401 | 235 | 636 |

**Table 10.** Percentage of achievement of the "Scientific Awakening" programme according to the milieu (urban/rural).

|  |  | **Environment** | | **Total** |
|---|---|---|---|---|
|  |  | **Urban** | **Rural** | |
| % of Achievement of the "Scientific Awakening" program | No answer | 29 | 10 | 39 |
| | 70% | 4 | 6 | 10 |
| | 80% | 28 | 24 | 52 |
| | 90% | 115 | 86 | 201 |
| | 100% | 225 (56.1%) | 109 (46.38%) | 334 |
| **Total** | | 401 | 235 | 636 |

**Table 11.** Use of the documents prepared by the teacher in the urban and rural milieu.

|  |  | **Environment** | | **Total** |
|---|---|---|---|---|
|  |  | **Urban** | **Rural** | |
| **Documents prepared by the teacher** | No answer | 9 | 5 | 14 |
| | Not Used | 10 | 7 | 17 |
| | Rarely | 31 | 18 | 49 |
| | Occasionally | 195 | 142 | 337 |
| | Often | 156 (38.9%) | 63 (26.8%) | 219 |
| **Total** | | 401 | 235 | 636 |

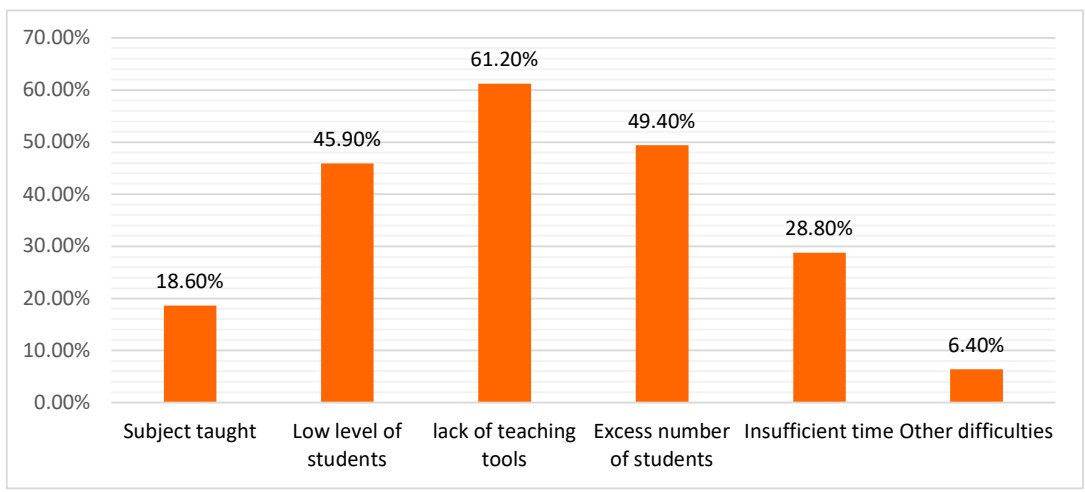

**Figure 4.** Different types of teaching difficulties encountered by teachers.

### 4.3. Correlations

The percentage of completion of the "Scientific Awakening" program significantly correlates with both certain teaching difficulties (Table 12) and with the teaching methods used (Table 13). Likewise, certain teaching methods and teaching difficulties correlate with each other (Table 14).

**Table 12.** Correlation between the rate of the "Scientific Awakening" program completion and teaching difficulties.

| | **Difficulties** | | **"Scientific Awakening" Programme** | **Student Level** | **Teaching Tools** | **Time Allowed** |
|---|---|---|---|---|---|---|
| Spearman's rho | % of completion of the "Scientific Awakening" -program | Correlation coefficient | 0.174 ** | 0.081 * | 0.128 ** | 0.099 * |
| | | Sig. (2-tailed) | 0.000 | 0.041 | 0.001 | 0.012 |
| | | *N* | 636 | 636 | 636 | 636 |

** Correlation is significant at the 0.01 level (2-tailed). * Correlation is significant at the 0.05 level (2-tailed).

**Table 13.** Correlation between % of "Scientific awakening" programme completion and methods used.

| | | | **Dogmatic Method** | **Interrogative Method** | **Active Dialogue** | **Discovery Method** | **Prior Preparation by the Students** | **Demonstrative Method** |
|---|---|---|---|---|---|---|---|---|
| Spearman's rho | % of completion of the "Scientific awakening" programme | Correlation coefficient | −0.113 ** | −0.099 * | 0.205 ** | 0.106 ** | 0.135 ** | 0.134 ** |
| | | Sig. (2-tailed) | 0.004 | 0.012 | 0.000 | 0.008 | 0.001 | 0.001 |
| | | *N* | 636 | 636 | 636 | 636 | 636 | 636 |

** Correlation is significant at the 0,01 level (2-tailed). * Correlation is significant at the 0,05 level (2-tailed).

**Table 14.** Correlation between methods and teaching difficulties.

| | | | Dogmatic Method | Interrogative Method | Active Dialogue | Group Work | Discovery Method | Prior Preparation by the Students | Demonstrative Method |
|---|---|---|---|---|---|---|---|---|---|
| Spearman's rho | Subject taught | Correlation coefficient | −0.088 * | −0.081 * | 0.053 | 0.042 | 0.159 ** | 0.163 ** | 0.018 |
| | | Sig. (2-tailed) | 0.027 | 0.040 | 0.180 | 0.287 | 0.000 | 0.000 | 0.652 |
| | | N | 636 | 636 | 636 | 636 | 636 | 636 | 636 |
| | Student level | Correlation coefficient | −0.066 | −0.180 ** | 0.125 ** | 0.158 ** | 0.205 ** | 0.067 | 0.076 |
| | | Sig. (2-tailed) | 0.095 | 0.000 | 0.002 | 0.000 | 0.000 | 0.092 | 0.057 |
| | | N | 636 | 636 | 636 | 636 | 636 | 636 | 636 |
| | Teaching tools | Correlation coefficient | −0.102 * | −0.201 ** | 0.125 ** | 0.111 ** | 0.149 ** | 0.086 * | 0.130 ** |
| | | Sig. (2-tailed) | 0.010 | 0.000 | 0.002 | 0.005 | 0.000 | 0.030 | 0.001 |
| | | N | 636 | 636 | 636 | 636 | 636 | 636 | 636 |
| | Excess number of students | Correlation coefficient | −0.037 | −0.073 | 0.029 | 0.129 ** | −0.045 | −0.070 | 0.029 |
| | | Sig. (2-tailed) | 0.348 | 0.067 | 0.460 | 0.001 | 0.254 | 0.079 | 0.470 |
| | | N | 636 | 636 | 636 | 636 | 636 | 636 | 636 |
| | Time allowed | Correlation coefficient | −0.085 * | −0.061 | 0.023 | 0.046 | 0.016 | 0.046 | 0.060 |
| | | Sig. (2-tailed) | 0.032 | 0.127 | 0.558 | 0.247 | 0.682 | 0.252 | 0.134 |
| | | N | 636 | 636 | 636 | 636 | 636 | 636 | 636 |

** Correlation is significant at the 0.01 level (2-tailed). * Correlation is significant at the 0.05 level (2-tailed).

## 5. Discussion

The plurality of types of objectives (cognitive, socio-affective, and behavioural) that can be targeted by environmental education programs leads to the inevitable adoption of a multitude of pedagogical approaches and an eclectic multi-methodological position [49,61,66,90,91].

In this study, the teachers appreciated various pedagogical approaches: affective-moral, religious, behavioural, cooperative, and problem-solving ones. However, we found no significant difference between the views of teachers in urban and rural milieus regarding their preferred approaches (Table 6). Similar to the teachers, several researchers have emphasized the importance of an affective approach in environmental education especially in programs that are built on the basis of students' meaningful environmental experiences (e.g., [58,62]). Littledyke [90] and Jeronen et al. [61] in their part also argue that the affective and cognitive domains must be explicitly integrated in environmental education because responsibility and a sense of relationship are essential for taking informed action and participating in environmental protection.

Similar to teachers, several researchers have valued and supported the religious approach to environmental education. Hitzhusen [64] stated in his research that religion is a useful subject, as it contains several perspectives that can enhance and supplement environmental education. He asserted that the thoughtful inclusion of religious elements can offer students a richer range of perspectives from which to examine environmental values. Parker [65] has argued that placing the environment within a religious cosmology has the potential to minimise cognitive dissonance in school learning and to enable the development of environmental religious ethics and practice, using values derived from religious teachings. Likewise, Crowe [63] argued that integrating spirituality and religious themes into environmental education is a way to connect students to their meaning systems. It allows students to have authentic learning experiences and to give sense to the knowledge they acquire in the classroom. It can be seen as an alternative pedagogical approach that supports the change of learners' environmental attitudes and behaviour [63].

Similar to some teachers, some researchers also stress on the pragmatic pedagogical approaches, such as the behavioural approach [66,67], the cooperative approach [68,69], and the problem-solving approach [70,71].

Fontes [92] indicated that action skills require EE to integrate distinct approaches and knowledge of different theoretical and practical types, in addition to the Willingness to act. However, the content and pedagogical approaches of environmental education vary according to the teachers' views and educational goals [93,94]. In general, the types of pedagogical approaches likely to be adopted are largely influenced by the target skills, the addressed environmental themes, the available means, the characteristics of the students, and the learning environment.

In this research, the teachers valued systemic, interdisciplinary, and critical approaches less than the above-mentioned ones. The result differs from previous results, according to which these approaches are highly valued by researchers specialised in environmental education. Several researchers and organisations have recommended that the systemic approach should be used in environmental education [32,95–102].

Regarding the systemic approach, the teaching of the environment as a system (ecosystems) does not begin in Morocco until secondary education. The negligence of this approach by Moroccan primary school teachers has been confirmed by another very recent research which has found that Moroccan primary school students struggle to grasp the meaning of the environment as a "system" [32]. This research recommended teaching the environment as a "system" from the fifth grade of primary education. This would help the students to take into account the complexity of the network of relationships between the components of the environment and to understand the interdependence and interactions of living beings with each other and with their living environments better than the current situation [32]. Certainly, we endorse this proposition in our research. In the same vein, among the key skills for sustainable development according to UNESCO, we find the skill of systemic thinking and the skill of critical thinking [19]. The critical thinking skill includes the ability to "question norms, practices and opinions; reflect on one's own values, perceptions and actions, and take a stand..." [18]. Furthermore, the teaching and learning of sustainable education must be based on values and on holistic, systemic, and interdisciplinary planning and implementation processes [18]. In addition, the interdisciplinary approach is also well appreciated in environmental education [21,36,101,103–105]. Likewise, the critical approach is well appreciated and recommended by the research community [22,26,97,98,104–110]. This approach values the practice of critical thinking with regard to our socio-environmental realities and choices. The discrepancies identified between the position of the teachers subject to the study and the position of the scientific community can be explained by certain objective and subjective reasons. First, there is no continuing education and any formal environmental education program in the Moroccan education system [86]. Therefore, teachers are generally not aware of the scientific issues concerning environmental education. Second, it appears that the three pedagogical approaches least valued by the teachers are also the least used. This is probably due to the teachers' humble familiarity with these pedagogical approaches and their insufficient training to implement such pedagogical approaches. Teachers also generally prefer to use their own teaching methods and pedagogical approaches independently of the work done by other teachers [111,112]. One reason may also be that, for example, an interdisciplinary approach requires additional effort, interdisciplinary cooperation, and planning [113].

The most widely used teaching method in the "Scientific Awakening" course was oral presentations by a teacher (dialogue and demonstrative method). The "demonstrative method" [114–116] is a teaching method in which "the teacher is the principal actor while the learners watch with the intention to act later. Here, the teacher does whatever the learners are expected to do at the end of the lesson by showing them how to do it and explaining the step-by-step process to them" (Quoted by Daluba, [115], (p. 2)).

The student-centred active methods (working in small groups, discovery learning) were scarcely used. The types of teaching tools used showed that experiments conducted

in the laboratory and ICT-assisted demonstration were also rarely used. (Figure 3). Audio-visual tools, ICT, laboratory equipment, and school field trips were almost completely missing. Teaching was based almost exclusively on textbooks. The results concerning the lack of use of active methods accompanied by an often-excessive use of the textbook as the only teaching tool have been confirmed by another recent research which has approached environmental education in the same region of Morocco (Fez-Meknes) [49]. We can say that these results are indicators of less effective learning. According to some recent studies, there is a significant correlation between the frequent use of several active methods, the rational diversification of teaching materials, and the increase in students' performance [49,73]. On the other hand, the excessive use of the textbook as the only teaching tool has given rise to several criticisms [49,117,118]. Some authors have even linked the students' poor performance to the dominance of the textbook as the only teaching material [49,119].

Teachers in urban areas prepare more teaching materials (38.90%) than their fellow teachers in rural areas (26.80%) (Table 11). It is concluded that teachers in urban areas are relatively more active than their peers in rural areas. This is confirmed by the rate of teachers who were able to fully complete the "Scientific Awakening" program (100%). Indeed, 56.1% of the teachers who managed to complete the program are located in urban areas while only 46.38% in rural areas (Table 9). This indicates that working in rural areas is more demanding. Admittedly, we know that mixed classes containing several grades taught at the same time by the same teacher are only found in rural areas. In addition, the majority of teachers in rural areas live in cities (urban areas), far from their place of work [86], which further complicates their task. Sometimes, the weather conditions (flooding especially) prevent students and their teachers from reaching their schools. We can say that the rural environment in Morocco requires even more attention from those in charge of the education sector by solving such chronic problems and, thus, ensuring equitable and quality education for all.

The used pedagogical methods and tools will not allow the students to achieve certain types of objectives, especially those related to action, savoir-faire, manipulation, and soft skills. Due to the almost complete absence of certain methods (working in small groups, discovery learning, field trips, experiments conducted in laboratory and ICT-assisted demonstrations), students' opportunities to work actively in the learning process and develop their assessment and interpersonal skills are limited. This inefficiency to achieve the educational objectives was confirmed by the percentage of teachers who were able to complete the entire "Scientific Awakening" program (52.5%) (Table 7). 47.5% of the teachers were unable to achieve all the objectives of finish this program. So, it is clear that the study revealed a real pedagogical problem. This problem is quite apparent given that 70.9% of the teachers reported having teaching difficulties in environmental topics assigned in the "Scientific Awakening" course (Table 8). Currently, it is widely recognized that teaching methods and practices directly affect students' performance and learning [120–123]. The completion rate of the "Scientific Awakening" program is positively and significantly correlated with certain types of teaching difficulties, especially with the teaching difficulties related to the content of this program, lack of teaching tools, insufficient time allocated to teaching, and poor knowledge level of the students (Table 12). This correlation seems paradoxical, but it is logical to a certain extent. In fact, the teachers who were most aware of the existence of these types of difficulties are the ones who managed to cover a higher percentage of the program. On the other hand, teachers who achieved a lower percentage of progress identified fewer teaching difficulties in their own pedagogical action. Given that the teachers worked in almost the same conditions (didactic tools, same time, same program), it can be said that the teachers who best perceived the problems of their pedagogical action were also able to solve them more effectively [124–126].

The completion rate of the "Scientific Awakening" program is also significantly correlated with the teaching methods used (Table 13). In fact, the completion rate of this program correlates negatively with the traditional transmissive methods (dogmatic and interrogative) and positively with the active methods (active dialogue, discovery learning,

prior student preparations and demonstrative method). This is the opposite of what some teachers may believe. For the reason that the traditional transmissive methods which seem the easiest to use were the least effective in achieving all the educational objectives. In other words, the teachers who used these traditional methods the most were the ones who completed a lower percentage of the program. On the other hand, the active methods, which are the most difficult to adopt, allowed for a more effective achievement of the educational objectives. The effectiveness of the teaching action of the teachers was measured quantitatively through the percentage of achievement of educational objectives (% of program completion) in addition to the qualitative measure relating to the type of teaching methods. The teachers concerned with using active methods were the ones who were able to achieve a higher percentage of the "Scientific Awakening" program when compared with their peers who used traditional teaching methods.

The teaching methods used and the teaching difficulties encountered also correlate with each other (Table 14). However, in general, teaching difficulties are positively correlated with the active teaching methods and negatively with traditional transmissive teaching methods. That is to say, the more the teachers used active teaching methods, the more teaching difficulties they encountered in their teaching action, but they carried out "Scientific Awakening" program more efficiently. On the other hand, the more the teachers used traditional teaching methods, the less teaching difficulties they encountered, but their action was less effective in achieving the educational objectives. Admittedly, the teaching impact is not obvious enough, and the presence of difficulties should not be considered as a negative sign, but rather a precursor for the promotion and positive evolution of the pedagogical action. It seems that the teachers who are most concerned about delivering quality work are the most demanding and the most aware of the existence of teaching difficulties and problems in their educational action. Indeed, the problems and constraints they face push them to find more creative and appropriate solutions to achieve their goals more effectively.

The correlations between teaching methods and difficulties can be analysed in detail according to their significance as follows: first, the difficulties relating to the lack of teaching tools constitute the main difficulty significantly correlated with all types of methods used (Table 14). Certainly, this problem maintains a very important position in the pedagogical action of teachers regardless of their pedagogical methods. This same problem (lack of educational tools) has been detected by other similar studies [49,73,86]. So, this problem surfaces in reality; however, most teachers who are aware of its existence are those who use active methods more in their teaching practice. This explains the existence of significant positive correlations between the use of active methods (active dialogue, group work, discovery method, demonstrative method) and the acknowledgment of this problem (Table 14). However, there are significant negative correlations between the declaration of this problem and the use of traditional methods (dogmatic and interrogative). In other words, teachers who use more traditional methods do not notice the existence of a problem relative to the lack of teaching tools. This is probably because they are used to methods that do not require enough pedagogical tools, and they do not think of using new pedagogical tools later. We can say that the use of educational tools and its importance gives us general indicators on the types of methods used (active or traditional). Several researchers support the positive influence of educational tools on the academic performance of learners [127–131]. ICT and technological tools, whose use by teachers is limited in our study sample, are widely recognised by many researchers as being very beneficial in facilitating learning and promoting pedagogical effectiveness at multiple levels [128–131]. Some studies have focused on the effectiveness of technological tools in the teaching-learning process and have made comparisons between teaching methods and the technological tools that correspond to them to conclude that teaching using technological tools has proven to be more effective than the traditional teaching methods [132].

The second teaching challenge, strongly correlated with the methods used, is the cognitive competency of the students. Indeed, teachers cannot work with all students



who show divergent performance with the same methods. However, the teachers who reported that there is a real problem relating to the inconsistent level of students are those who use more active methods, particularly, the discovery method, small group work, and active dialogue, while the interrogative method is not often used (question-answer). This significant negative correlation between the problem of unsatisfactory students' level and the use of the interrogative method seems logical to a certain extent. In fact, the weaker the level of knowledge acquired by students is, the less they tend to interact with their teacher's questions. So, the teacher would eventually ask fewer questions. On the other hand, the better the cognitive level of the learners is, the better answers the teacher would get, i.e., the interaction would reach, thus, optimal levels.

For the teaching difficulties relating to the subject taught, they significantly and positively correlate with the method of discovery and with the prior preparations of the students. It seems that the teachers who find the subject matter difficult are the ones who provide more opportunities for students to do research, pre-lesson preparation, and self-discovery of the learning topic, which can make it easier for them in the classroom.

Concerning the challenges relating to the large-number classrooms, which is reported by half the surveyed teachers (49.4%), the challenge is significantly correlated with only the group work method. This seems to be very logical since teachers who complain about the large number of pupils per class tend to prefer to work with the method of small groups. This method seems effective according to several researchers [133–136]. Some authors [136] have identified several advantages relative to the use of this method. For example, we can cite the mastery of educational content, the development of communication skills between students, saving time and opportunities for self-assessment.

Finally, the least significant or almost non-existent correlation of methods was found with the time factor (given to the "Scientific Awakening" programme) (Table 14). Moreover, given that the dogmatic method is rarely used (Figure 2), we can say that the insignificant time allocated to this programme does not remarkably correlate with any teaching method used. This seems logical given that the time allotted to the "Scientific Awakening" programme is a fixed transversal factor and given equally to all teachers regardless of their methods. We can conclude that teachers adapt their methods with the time allotted and dedicate more importance to respecting time to the detriment of the type of methods.

In general, we can say, at the end of this analysis of the correlations (methods/difficulties), that the detection and identification of pedagogical problems and difficulties constitute the first step toward any readjustment and positive evolution in the pedagogical action of teachers, particularly the pedagogical methods likely to be adopted. The process of pedagogical development initiated by the detection of teaching difficulties can be described as shown in Figure 5 (inspired by Dai et al. [137]).

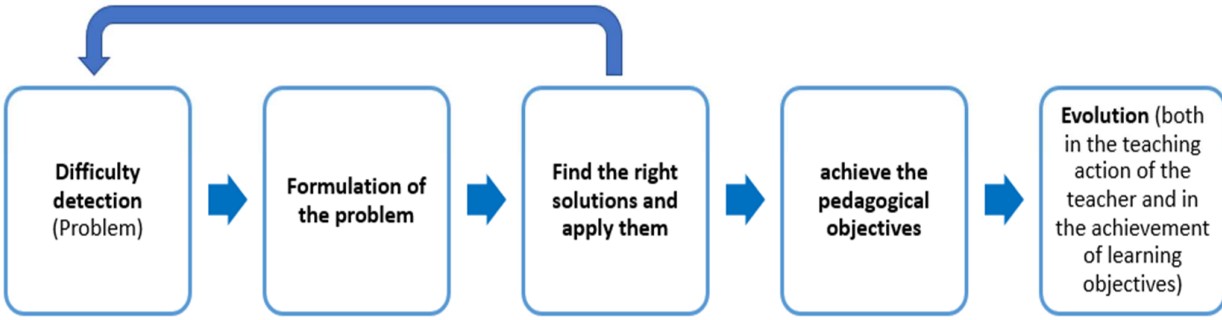

**Figure 5.** Simplified diagram of the main stages of a positive development in the educational action of teachers (inspired by Dai et al. [137]).

Based on our professional experience (as teaching researchers, teacher trainers and former practising teachers) we can argue that the teachers who demonstrate effective classroom management are able to detect even small problems and disruptive factors and to solve them early effectively, and it is the same for teaching and learning problems. The

detection of these problems and the awareness of their importance and impact constitute the essential condition and the starting point of any readjustment and any positive evolution in the pedagogical action of teachers [124–126,138,139].

## 6. Research Limits

Even though we worked with an exhaustive sample of the region of Fez-Meknes, it is yet to confirm (or invalidate) the results obtained with further research extending to other regions and other countries. On the other hand, the lack of pedagogical effectiveness was measured based on the percentage of completion of the "Scientific Awakening" syllabus. However, other research can measure the effectiveness of teachers' pedagogical action by referring to the academic performance of learners. This may confirm or invalidate some of our results. Finally, we proposed, at the end of this study, a continuous training for the benefit of primary school teachers, precisely about the systemic and interdisciplinary approach but, currently, nothing is known about the impact of such training on the effectiveness of the pedagogical action of teachers.

## 7. Conclusions

The importance of using different pedagogical tools provided us with general indicators for the main classes of pedagogical methods used (active or traditional). Moreover, taking into account the frequency of completion of the "Scientific Awakening" program and the frequency of production of educational materials, we found that these indicators of the lack of educational effectiveness are more reported in rural areas. Therefore, the rural environment requires more attention and work from those in charge of the education sector in Morocco if we are to ensure quality and equitable education.

Considering the results obtained, and to improve the pedagogical action of the teachers, we recommend conducting continuous training for the benefit of teachers. This ongoing training must focus on several areas in particular, the clarification, the concretization, or even the implementation of certain approaches to education relative to the environment (the systemic and interdisciplinary approach). The process should be undergone in parallel with training on the effective use of active teaching methods, as well as the rational and creative diversification of teaching tools. This continuous training must necessarily be based on the needs of a teacher whose commitment is crucial to the completion of the "Scientific Awakening" syllabus and therefore the achievement of all the educational objectives. Such training will allow, among other things, the exchange of experiences between colleagues and sharing of the most successful innovative experiences. This can only be beneficial to improve and promote the pedagogical action of teaching.

It should be noted that the study benefited from a measure of reliability and external and internal validity. Reliability and external validity were expressed by the completeness of the sample with which we worked (636 teachers from the Fez-Meknes region) and the tested reproduction of the results. In addition, the relevance of the results, concerning the teaching methods and tools used, has been confirmed by other recent studies conducted in the same region of Morocco [49]. Internal validity manifests in the close links and mutual confirmations between the measured variables (construct validation), particularly the modest use of active methods combined with the inadequacy of the teaching tools used. This inadequacy was confirmed by the types of teaching difficulties encountered and the resulting lack of pedagogical effectiveness. Finally, the study revealed that being aware of the existence of pedagogical problems constitutes the starting point for any positive development in the pedagogical action of teachers.

**Author Contributions:** Conceptualization, B.E.B. and L.M.; Methodology, B.E.B., E.J. and L.M.; Validation, E.J. and M.L.; Formal analysis, B.E.B., L.M. and J.K.; Data curation, B.E.B., J.K. and J.I.; Writing—original draft preparation, B.E.B.; Writing—review & editing, B.E.B., E.J., L.M., J.K., J.I., A.A. and M.L.; Supervision, B.E.B. and L.M.; Funding acquisition, B.E.B., L.M., E.J., J.K., J.I., A.A. and M.L.; All authors have read and agreed to the published version of the manuscript.

**Funding:** This research received no external funding.

**Institutional Review Board Statement:** This study followed the ethical principles of the Declaration of Helsinki in terms of confidentiality, anonymity and use of information for research purposes only.

**Informed Consent Statement:** Informed consent was obtained from all subjects involved in the study.

**Data Availability Statement:** Not applicable.

**Conflicts of Interest:** The authors declare no conflict of interest.

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
