# Peer review of "Teaching Environmental Themes within the “Scientific Awakening” Course in Moroccan Primary School: Approaches, Methods and Difficulties"

_education, doi:10.3390/educsci12110837_

Round 1

Reviewer 1 Report (Previous Reviewer 1)

Dear Authors,

-The line of the research and the objectives of the study are clearly identified

-The discussion is more efficient and responds to the results

-The results have taken into account the objectives of the research

-The theoretical contribution of work and the conclusions could be improved and relevant

-More contemporary Literature should be added

Author Response

Reviewer 2 Report (Previous Reviewer 3)

Dear Colleagues,

First of all, thank you for your response to the comments previously made. 

Secondly, I wanted to let you know that your manuscript has improved considerably. It is much clearer and conveys the information obtained with a high degree of scientific rigor.

Sincerely, 

Author Response

Reviewer 3 Report (Previous Reviewer 2)

Dear Author/s,

I have read with attention your paper. It could surely be interesting and stimulating from different points of view but also after its re-arrangement, I continue to see important problems.

I wish to share with you here below, only the most important of them.

a)     The framework: You declared that it is  EE but:

Lines 31-75 in spite of your speech about EE in which you underlined the importance of attitudes, values and behaviors, your paper is related only to knowledge in a very simple way (disciplinary knowledge and not interdisciplinary ones!).

Line 409-410 – The skills discussed by UNESCO are relate to Education for Sustainable Development and not to EE.

Lines 413-414 – You mentioned sustainable education.

I’m sorry but EE, Education for SD and sustainable education are not synonyms.

Finally, (I’m sorry but…) please, let me underline that from line 444 to the end of the paper (line 642) the term “EE” is no more considered. It seem to me quite strange in consideration that the paper aims to be considered in this framework.

b) The title: You titled your paper “…impact of pedagogical methods on teaching effectiveness” but you haven’t collected data on “teaching effectiveness”. Your references (lines 62 and line 70 in the old version and lines 113-120) seem to me quite poor and simplistic to legitimate your words “are good indicators of effective learning”. In what sense? In relation to knowledge? Or something else related to EE (values, attitudes, behaviors, for instance)?

c) The methodological rigor: Very frequently, in your paper you discuss about data/results that are not shown or you mentioned aspects not clearly shown.

Some examples of this:

Lines 202- 205 – These data are not represented in the tables mentioned above (n.1, 2 or 3)

Lines 224 - You mentioned “this second part of the questionnaire” but you didn't give information about it.

Lines 236-240 – You have written “before giving choices…” but where? In the same questionnaire? It’s not clear how have you operated.

Lines 329-333 - These data are not shown in the figures (tab.6?)

Line 339 – Caption of the figure is unclear: is  “number of teachers who encountered the different difficulties” or “percentage of teachers that…”

Lines 362-363 – Where this data is shown?

Lines 453-458 – Where are shown these results shown? I cannot see evidence of these in your paper.

Round 2

Reviewer 1 Report (Previous Reviewer 1)

Τhe manuscript is well organized and structured. The theoritical contribution of the work as well as the conclusions are relevant. Its research purpose is a clearly stated and an apropriate method of study is applied. The results are clearly and there is cohensive connection beteen conclusions and  discussion

I suggested that the paper should be accepted after taking into account the following remark:

Some bibliography resourses should be replaced by contemporary Literature

This manuscript is a resubmission of an earlier submission. The following is a list of the peer review reports and author responses from that submission.

Round 1

Reviewer 1 Report

The manuscript is well organized and structured. Also propsect for further research is provided.  However, there should be a more cohesive between  conclusions and discussions. The bibliography should be more contemporary.

Reviewer 2 Report

Dear Author/s,

I have read with attention your paper that in my opinion could surely be interesting and stimulating from different points of view. In spite of this, in my opinion it needs a complete and depth re-thought and re-arrangement:

a)     From a cultural/scientific perspective;

- You focused on Moroccan primary school teachers but you didn't mention their characteristics (for instance: how is their pre-service and in service training? And their training on environmental issues?)

- You titled your paper “…impact of pedagogical methods on teaching effectiveness” but you haven’t collected data on “teaching effectiveness”. In fact:  you haven’t not considered students at all;  you haven’t analyzed the “effects” of the methods used by teachers and you haven’t collected  evidences of teaching effectiveness (for instance students evaluations, exams, assignments, and other materials). So, how can you speak about “teaching effectiveness”? I’m sorry but I’m not able to understand your approach to this point. Your references (lines 62 and line 70) seem to me quite poor.

- You mentioned “environmental education” but it’s not clear in what framework and with what international references you consider it. Your references are often quite dated (before 2000) and you didn't mention at all the debate in the field of EE. What is EE? (In line 75 you wrote “EE can be seen as lifelong learning”!) Is it “education about/in or for the environment? In your paper it seems to me that you consider EE as “education about the environment” or synonym of “science education”. In any case, you need to explicit and discuss this aspect and this debate need to be part of the “introduction” section.

- You wrote that “The study showed that Moroccan primary school teachers need in-service training for the adoption of systemic and interdisciplinary pedagogical approach” but it seems to me that you have not sufficiently focused your attention on these aspects.

b)     From a methodological perspective;

-  You mentioned the “scientific awakening” course. It’s sound quite strange this term; do you mean “scientific awareness”?. In any case, you haven’t given information about the course. How many hours are there weekly? What topics does it cover (in addition to LES)? These information are fundamental to understand teachers’ feelings and ideas (and to understand lines 245 and later);

-  The description of your “data collection tool” is quite imprecise and partial.

For instance:

Line 145 - you mentioned the teachers opinions but the responses you showed are not coherent with this; they seemed related to the frequency in using the different methods.

Line 152 -you mentioned the second part of the questionnaire but you didn't give information about it

Line 156 – you mentioned that “teaching difficulties usually encountered by teachers “ were mapped. You have written that “have mapped teaching difficulties relate to …” (see lines 158 and later) but it’s not clear how you have operated.

Line 167 – You wrote “to characterize the opinions of the teachers on each type of pedagogical approach, the five-point Likert scale was used (0 points to "no answer", 1 point to "disagree", 2 points to "indifferent", 3 points to "somewhat agree" and 4 points to "strongly agree").” How have you transformed “an opinion” in one of these different positions? (disagree, indifferent and others?). It’s impossible to understand, if you don’t give the possibility to read the exact formulation of the questions in the questionnaire.

- Your second research question is: “What is the relationship between the teaching methods used and the teaching difficulties encountered by teachers in their teaching of the "Scientific Awakening" -program?” It seems to me that the question at first need to be: “are there relationship between….?” And only if your data confirm this, the question could be also “what is the relationship….?”

- You have separated “urban” by “rural areas” (you have written that “The educational and socioeconomic differences existing between the two environments have been considered also in this study). In spite of this,  the results are not discussed considering these two different contexts (you mentioned only overcrowding – lines 250-251). If you are not interested in explain and discuss the differences between contexts the distinction of the two sub-samples, become completely unuseful.

- You have written as in lines 201 -206. Why you have done these choices? In this way you don’t pay attention to methods and tools rarely used by teachers but that could be very interesting and stimulating.

- The discussion section is confusing. Without a clear idea of the questionnaire submitted to the teachers,  it’s difficult to understand if your results are coherently discussed. In addition some aspects, in my opinion, should be better approached in the introduction section.

These are only some of the most important weaknesses of the paper, in my opinion. For this, I’m sorry, but I suggest a complete re-thought and re-arrangement of the paper before a new submission to an international educational journal.

Finally, I’m going to suggest also a linguistic revision of your manuscript by a native English speaker; I think that probably some mistakes and misinterpretations of your work could be related to an improper use of different terms.

Thank you for your attention!

My best regards

Reviewer 3 Report

Dear colleagues,

I believe that the work provides relevant results to increase knowledge of the teaching-learning system in Morocco nowadays. 

However, an excess of references  is observed in the manuscript (103), many of them are excessively old when used to describe a current system in a cross-sectional study in which its historical evolution is not evaluated, for this reason I consider it can be rewritten by selecting the sources and adding some, such as line 383, in which a reference would be necessary to confirm the statement made.

In addition, I consider that specifically Table 10 of the manuscript contains a lot of information, but it is little discussed, I think it should be emphasized and a deeper discussion of that part in particular should be made.

Reviewer 4 Report

Considerations to the article:

1. Article structured around the main objective pursued, with an appropriate title and a summary that illustrates the objective, procedures and results.

2. In the introduction, the line of research is clearly identified, although there is a deficient work on the antecedent literature. The objectives to be achieved from the work are clearly appreciated.

3. In the methodological field, a series of deficiencies are detected: when the participants are described, there is no data, in the sampling, on the population from which it has been referenced. This causes the absence of data on sampling error, which prevents a generalization of the results.

4. There is a specific definition of the variables studied and clarity in the procedure, although the instruments should be more clearly described.

5. Their reliability has an inadequate level, for which the Chrombach coefficient should be taken into account and observed.

 6. The statistical procedure used, although informative, is not very useful to conclude on results that can be taken into account. Perhaps it would be more appropriate to carry out differential analysis.

 7. The results end up being poor considering the objectives of the work.

8. The discussion is deficient and responds with the poverty and deficit expressed in the results. The contributions regarding the background research are very elementary, if not circumstantial or anecdotal. Without forgetting that these conclusions are mostly more results. Their clarifications and comments do not coincide or contradict criteria expressed in other investigations and authors who clarify and carry out similar investigations.

 9. What has been stated leads us to determine that the theoretical contributions of the work, as well as the conclusions, are not very relevant.